# Planting Date and Hybrid Affect Sugarcane Aphid Infestation, Yield, and Water Use Efficiency in Dryland Grain Sorghum

**Zane Jenkins** [1,2], **Sushil Thapa** [2,3], **Jourdan M. Bell** [2], **Kirk E. Jessup** [2], **Brock C. Blaser** [1], **Bob A. Stewart** [1] and **Qingwu Xue** [2,*]

1   Department of Agricultural Sciences, West Texas A&M University, Canyon, TX 79015, USA
2   Texas A&M AgriLife Research and Extension Center, 6500 Amarillo Blvd. W., Amarillo, TX 79106, USA
3   Department of Agriculture, University of Central Missouri, Warrensburg, MO 64093, USA
*   Correspondence: qxue@ag.tamu.edu; Tel.: +1-806-354-5803

**Abstract:** Grain sorghum (*Sorghum bicolor* L.) is a major dryland crop in the Texas High Plains. Currently, drought and infestation by the sugarcane aphid (SCA, *Melanaphis sacchari*) are the two major challenges to grain sorghum production in the area. A 2-year field study was conducted to investigate the effect of planting date (PD) and hybrid selection on yield, evapotranspiration (ET), water use efficiency (WUE), and SCA infestation. Five sorghum hybrids (86P20, SP-31A15, AG1201, AG1203, and DKS37-07) were grown on two planting dates (PD1—early May; PD2—late June) under dryland conditions. Insecticides were not used. There were significant differences in grain yield, WUE, evapotranspiration (ET), and SCA population between two PDs and among hybrids. For PD1, SCA infestation occurred after sorghum reached physiological maturity in 2017. Although SCA infestation was observed during late grain filling in 2018, SCA populations were low and did not affect yield. For PD2, SCA was present before anthesis in both years and significantly affected grain yield. Even with heavy SCA infestation in PD2, the grain yield was higher in PD2 than in PD1 due to timely precipitation. Among hybrids, AG1203, 86P20 and DK37-07 performed better with higher yield and less SCA infestation in PD2. Grain yield was more related to seeds per plant than to kernel weight and harvest index.

**Keywords:** dryland farming; evapotranspiration; *Melanaphis sacchari*; water use efficiency



## 1. Introduction

Grain sorghum is the third most produced grain crop in the U.S. and the fifth most produced grain crop in the world [1]. Grain sorghum is a drought-tolerant crop that performs well under the limited water [2–5]. Consequently, most U.S. grain sorghum production is in the dryland areas of the Great Plains, from South Dakota to Texas [6]. Across the Texas High Plains (THP), irrigation from the Ogallala Aquifer is declining, causing producers to refocus on dryland crop production. As the growth of irrigated cropland slows, dryland agriculture will become more important to meet the world's growing food demands [7,8]. The Food and Agriculture Organization of the United Nations (FAO) estimated that 70% more food and fiber will need to be produced by 2050 [9]. While drought is a worldwide problem, the impact of drought is severe in dryland farming areas of the Southern Great Plains. Across this region, normal in-season precipitation is less than the crop potential evapotranspiration (PET), and drought frequently results in crop failure [10]. Combining abiotic water stress with biotic pressure can magnify crop losses.

The sugarcane aphid (SCA) has become a key biotic stress in recent years [11]. The insect was identified as a sorghum pest as early as 1958 in Africa [12], but it was not identified in the THP until 2014 [11,13]. Sugarcane aphids cause yield loss [11,12], reduced grain quality [14], harvest delays, and damage to harvest equipment due to honeydew [11,15,16]. The SCA creates honeydew, which supports the growth of sooty mold fungi that negatively affects

the sorghum development [15,16], and in some cases, complete crop failure [16]. In Texas, yield losses from SCA range from 30% to 100%, depending on year and location [11,15,16].

Traditionally, dryland planting dates are based on soil moisture, air temperature, soil temperature, and upcoming weather predictions [17]. The first historical recommended planting date for grain sorghum in the Texas High Plains was between May 15th and June 15th [18]. However, early grain sorghum planting date recommendations were closer to May 15th than June 15th, depending on the cultivar being grown [19]. As time progressed, the time frame of late May to the middle of June continued to be standard practice in the area [20,21]. Modifying the practice of planting dates is a strategy for producers to reduce production risk caused by drought and SCA infestation. Strategic planting date modifications may ensure coordination between periods of high water demand (anthesis to grain filling) and the traditionally high precipitation periods of the year for the region [6,22,23], but hybrid maturity length is an important consideration. Early maturity hybrids can reach physiological maturity before the onset of precipitation if planted early in the season [24]. Consequently, producers must coordinate hybrid maturity with the targeted planting date. In contrast, some other hybrids can endure the dryer periods in vegetative stages and then reach reproductive stages during traditional higher precipitation months [24,25]. Planting date modifications are also an effective tool to manage insect pests in grain sorghum [26–28]. We hypothesized that SCA infestation could be minimized using an appropriate planting date and hybrid. The objective of the study was to investigate the effect of different planting dates and hybrid selection on grain yield and reducing the effect of drought and SCA infestation under dryland production in the THP.

## 2. Materials and Methods

### 2.1. Experimental Design

A 2-year (2017 and 2018) field study was conducted at the Texas A&M AgriLife Research Field Laboratory at Bushland, TX (Lat. 35.19° N, Long. 102.06° W; elevation 1170 m). The soil is Pullman Clay Loam (fine, mixed, superactive, thermic Torrertic Pauleustoll). Treatments included two planting dates (early, PD1 and late, PD2) and five hybrids (AG1201, AG1203, 86P20, SP31A15, DKS37-07). Hybrids were selected based on their respective seed companies' performance for yield potential and drought tolerance. Among the hybrids, AG1201 and SP31A15 are in the early maturity group (55–60 days to flowering), and the other three hybrids (AG1203, 86P20, and DKS37-07) are in the early to medium maturity group (62–64 days to flowering). There is no SCA resistance gene in these hybrids; all hybrids have could be tolerant to SCA at some degree based on the field observations from the United Checkoff Program [29]. A split-plot design was used with PD as a main plot and hybrid as a sub-plot. The study was comprised of four replications in 2017 and six replications in 2018. The more replications in 2018 were mainly due to variability in natural SCA infestation. The year factor was considered as random due to the differences in temperature and precipitation in the two years.

Plots were grown in a winter wheat (*Triticum aestivum* L.)–fallow–sorghum rotation. Fields were prepared by incorporating previous crop residue using a disc plow, leveled using a sweep plow, and then bedded using a disk bedder, with the sorghum being planted on beds with 0.76 m row spacing. Weeds were managed as needed with mechanical and chemical methods. Plots were 3.05 m wide by 9.14 m long and planted at a density of 74,100 seeds ha$^{-1}$. Buffer plots were planted around the experimental plots to prevent edge effects. The fertility requirement was assessed by taking soil samples to a 0.9 m depth from three locations within the field before planting in both years. Based on the soil tests, fertilizers were applied to meet a yield goal of about 4.0 Mg ha$^{-1}$, which is an average dryland sorghum yield in the THP. In this research, early planting dates (PD1) were 5 May 2017 and 15 May 2018, and late planting dates (PD2) were 27 June 2017 and 18 June 2018. The differences in planting dates between two years were related to soil moisture conditions and field operation. For a dryland cropping system, crop planting is largely determined under adequate soil moisture conditions, which generally is occurring

after rainfall events. Therefore, the differences in rainfall events in different years can cause difference in the planting dates. In 2017, plots received only the natural precipitation during the growing season. Due to severe drought conditions at planting in 2018, plots were irrigated before planting (79 mm for PD1 and 83 mm for PD2) to ensure seed germination and crop establishment. The irrigation water was applied through gated pipes, and the amount was measured by a water meter. No irrigation water was applied in the rest of growing season from after planting to maturity.

### 2.2. Data Collection

Sugarcane aphid populations (number of aphids per leaf) were determined by scouting plants within the plots weekly using methods outlined by [30]. The average SCA population per plot was calculated by counting SCA colonies on five individual plants per plot. In 2017, SCA infestation occurred after sorghum reached maturity in PD1. As such, no SCA population was collected for PD1. For sorghum plants in PD2, SCA populations were found starting early September, and SCA populations were counted on 16 and 23 September 2017. In 2018, SCA infestation started about in the middle of July. For sorghum plants in PD1, SCA populations were counted weekly starting 13 July 2018 for 4 weeks (13 July, 19 July, 10 August and 17 August). For plants in PD2, SCA populations were counted starting 31 July 2018 for 9 weeks (13 July–27 September 2018). For both years, no foliar insecticides were applied to control SCA in this experiment.

Soil volumetric water content (SWC) was measured at planting and after harvest from each plot using the gravimetric method from the 0 to 1.2 m profile (0–0.15, 0.15–0.30, 0.30–0.60, 0.60–0.90, and 0.90–1.20 m). Soil cores were collected using a tractor-mounted Giddings hydraulic soil sampling machine. Soil samples were weighed and then oven dried at 105 °C for 48 h before obtaining oven dried weight. The seasonal evapotranspiration (ET) was calculated using a soil water balance method, and water use efficiency (WUE) was calculated as a ratio of grain yield to ET [31–33] as:

$$ET = SW_{pl} - SW_{ma} + P + I - R \tag{1}$$

$$WUE = Yield/ET \tag{2}$$

where $SW_{pl}$ and $SW_{ma}$ were soil water (mm) in 0–1.2 m profile at planting and maturity. P, I and R were amounts of precipitation, irrigation, and runoff between planting and maturity. The values of I and R in this study were 0, because there was no irrigation during the growing season, and field plots did not have runoff.

Plots were harvested at physiological maturity. Grain yield was determined by hand harvesting of 0.76 m by 3.05 m from the middle row from each plot. A subsample was weighed, oven dried at 130 °C for 18 h, and re-weighed to determine the moisture content of the seed. Grain yield samples were corrected to 14% moisture. Seed weight, thousand kernel weight (TKW), and plant height were measured from five plants per plot, which were the same five plants used to record SCA populations over the course of the growing season. Stover samples were oven dried at 60 °C to constant weight and then weighed to calculate harvest index (HI; the ratio of total dry grain weight to total dry aboveground biomass).

The daily weather data were from a NOAA weather station at Bushland, TX (http//ncdc. noaa.gov, accessed on 15 August 2022).

### 2.3. Statistical Analysis

Analysis of variance (ANOVA) was conducted in SAS 9.4 (SAS Institute, Cary, NC, USA) using a PROC MIXED procedure for the split plot design [34]. Effects were considered significant when $p \leq 0.05$ according to the LSD test. The planting date and hybrid were considered fixed effects, while replication was considered a random effect. Significant interactions were considered before analyzing the main effect factors. SCA population data were analyzed using PROC GLM with each day (when the population was counted) being analyzed separately. Regression models were developed and analyzed using PROC

REG in SAS 9.4., and the linear or quadratic functions were determined by $R^2$ and RMSE. Before conducting ANOVA and regression, the normality test was conducted using PROC UNIVERIATE model where a Shapiro–Wilk test was considered to verify that data were normally distributed. Furthermore, the homogeneity of variances was verified using the Hovtest Welch option for the PROC GLM model.

## 3. Results

### 3.1. Weather Conditions

Temperature and precipitation differed between the 2017 and 2018 growing seasons (Figure 1). In 2017, the maximum temperature was higher than the 30-year average from May–July, and it was close to the 30-year average in August and September. However, the maximum temperature in 2018 was greater than the 30-year average in every month during the growing season (May–October). The minimum temperature in the 2017 growing season was similar to the 30-year average except for October. In 2018, the minimum temperature was greater than the 30-year average during the growing season except for August. Comparing the 30-year average, there was less precipitation in May and June in 2017. However, there was more precipitation in July and August than the 30-year average. The 2018 sorghum growing season received much less precipitation from May to September as compared to 2017 and the 30-year average. There was 206 mm of precipitation in October 2018. However, the October precipitation was received after physiological maturity, so it was not beneficial to sorghum growth and development.

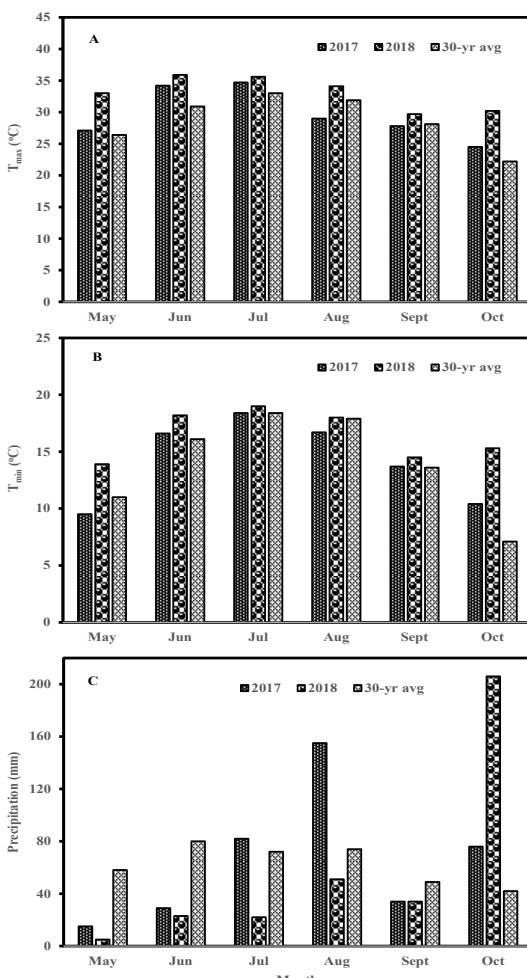

**Figure 1.** Mean monthly maximum temperature (T$_{max}$, **A**) and minimum temperature (T$_{min}$, **B**), and total precipitation (**C**) during sorghum growing season (May–October) and 30-year average (1981–2010) at Bushland, Texas (source: http//ncdc.noaa.gov, accessed on 15 August 2022).

### 3.2. Sugarcane Aphid Infestations

SCA do not over-winter on the Texas High Plains; rather, they migrate north each year from south Texas. In 2017, SCA was not found in PD1 plots during the whole growing season because sorghum had reached physiological maturity before SCA arrival. However, SCA was identified in PD2 plots and reached peak populations of about 1100 aphids per leaf in sorghum plants at 80 days after planting (DAP) (Figure 2). In 2018, SCAs were found in sorghum plants for both planting dates, but SCA populations in PD1 were much lower than SCA populations in PD2 (Figure 3). The greatest SCA populations were found at 88 DAP in PD1 on 10 August 2018 with approximately 168 SCAs per leaf, after which aphid populations decreased significantly. In PD2, the highest SCA population was recorded at 68 DAP on 23 August 2018, and the mean was 681 SCAs per leaf (Figure 3). SCA populations differed among hybrids (Figures 1 and 2). In 2017 PD2, the SCA population was measured only two times, at 80 and 87 DAPs. At 80 DAP, the hybrid DKS37-07 had the highest number of SCA per leaf, and SP31A15 had the lowest number. In 2018, SP31A15 planted on both planting dates had the highest number of SCA population at both rating periods. Of significance, SCA populations were estimated, but SCA damage ratings were not conducted.

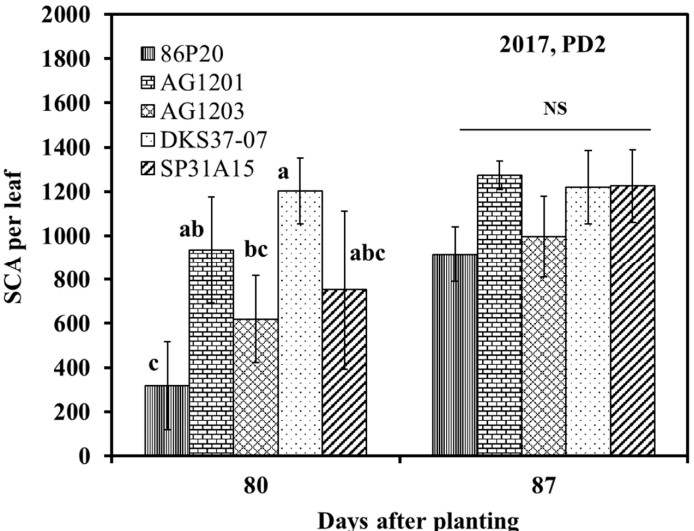

**Figure 2.** Average sugarcane aphid populations in each hybrid for 2017, PD2 grain sorghum. For 80 days after planting, means with the same letters are not significant at $p = 0.05$. NS: not significant ($p > 0.05$).

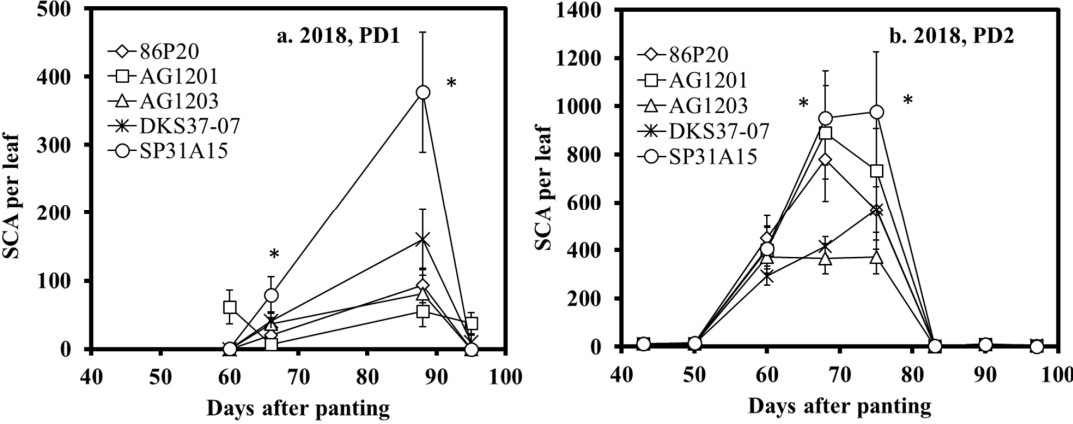

**Figure 3.** Average sugarcane aphid (SCA) population per leaf in each hybrid observed for the PD1 (**a**) and PD2 (**b**) grain sorghum in 2018. * Indicates the significant differences ($p < 0.05$) in SCA per leaf among hybrids.

### 3.3. Grain Yield

Grain yield was significantly affected ($p < 0.05$) by all main effects (year, Y; planting date, PD; and hybrid, H) and two-way interactions (Y × PD, Y × H, and PD × H) (Table 1). In 2017, the yields were greatest for PD2 with 86P20 (3758 kg ha$^{-1}$), AG1203 (3359 kg ha$^{-1}$), and DKS37-07 (3069 kg ha$^{-1}$) yielding significantly higher than other hybrids (Table 2). AG1203 (4173 kg ha$^{-1}$) and DKS37-07 (3703 kg ha$^{-1}$) were the highest-yielding hybrids in 2018. Across the two years, there was no hybrid difference ($p > 0.05$) in grain yield in PD1, while in PD2, AG1201 (4858 kg ha$^{-1}$), DKS37-07 (4493 kg ha$^{-1}$) and 86P20 (4420 kg ha$^{-1}$) had significantly higher yield than AG1203 (3486 kg ha$^{-1}$) and SP31A15 (3062 kg ha$^{-1}$) (Table 3). Comparing the two years, sorghum had a higher grain yield (3475 kg ha$^{-1}$) in 2018 season than in 2017 (3027 kg ha$^{-1}$), which was likely due to the result of different weather conditions between these two growing seasons (Figure 1 and Table 2). Of significance, across both years and all hybrids, the average grain yield was greater for PD2 (4045 kg ha$^{-1}$) than for PD1 (2279 kg ha$^{-1}$) (Table 3).

### 3.4. Harvest Index and Yield Components

A significant ($p < 0.05$) Y × H interaction was observed for harvest index and seeds per plant. The interaction was not significant ($p > 0.05$) for thousand kernel weight (Table 1). The Y × PD and PD × H interactions were observed for all three variables (harvest index, seeds per plant and thousand kernel weight). Harvest index, thousand kernel weight, and seeds per plant were different between the two years (Table 1). Planting date had a significant effect on these variables except for thousand kernel weight ($p = 0.248$). Across both years, harvest index ranged from 0.37 to 0.44 in PD1 and from 0.44 to 0.50 in PD2. Across the planting dates in 2017, there were no differences in harvest index between hybrids. However, 86P20 (0.53) and AG1201 (0.54) had higher numerical harvest indices than other hybrids. There was no hybrid difference in thousand kernel weight in 2017, while in 2018, hybrid AG1203 (20.92 g) had a lower thousand kernel weight than other hybrids (22.48–23.33 g). Between planting dates, AG1203 (18.0 g) in PD1 and AG1201 (18.2 g) in PD2 had the lowest thousand kernel weight. Across years, there was no significant difference seeds per plant among hybrids in both PD1 and PD2, while AG1203 had the highest number of seeds per plant in both 2017 (4352 seeds plant$^{-1}$) and 2018 (3272 seeds plant$^{-1}$) in PD1. Comparing the two years, sorghum plants had more seeds per plant in 2017 but had a greater thousand kernel weight and harvest index in 2018 (Table 2). In case of planting dates, plants in PD2 had a higher harvest index and seeds per plant, but there was no difference in thousand kernel weight between the two planting dates (Table 3).

**Table 1.** ANOVA (*p*-values) for grain yield, harvest index, thousand kernel weight (TKW), seeds per plant, evapotranspiration (ET), and water use efficiency (WUE).

| Effect | Grain Yield | Harvest Index | TKW | Seeds Per Plant | ET | WUE |
|---|---|---|---|---|---|---|
| Year (Y) | 0.0348 | <0.0001 | <0.0001 | 0.0055 | 0.0005 | 0.0375 |
| Planting date (PD) | <0.0001 | <0.0001 | 0.2480 | <0.0001 | <0.0001 | <0.0001 |
| Hybrid (H) | <0.0003 | 0.2165 | 0.0606 | 0.0030 | 0.9030 | 0.0016 |
| Y × PD | <0.0001 | <0.0001 | 0.0451 | <0.0001 | <0.0001 | 0.0005 |
| Y × H | 0.1088 | 0.0559 | 0.7493 | 0.0525 | 0.8420 | 0.1237 |
| PD × H | 0.0341 | 0.0932 | 0.0091 | 0.1142 | 0.3851 | 0.2143 |
| Y × PD × H | 0.1082 | 0.5235 | 0.2633 | 0.4988 | 0.3140 | 0.1545 |

**Table 2.** Means of grain yield, harvest index, thousand kernel weight (TKW), seeds per plant, evapotranspiration (ET), and water use efficiency (WUE) as affected by year and hybrid interaction.

| Effect | Grain Yield kg ha$^{-1}$ | Harvest Index | TKW g | Seeds Per Plant | ET mm | WUE kg m$^{-3}$ |
|---|---|---|---|---|---|---|
| **2017** | | | | | | |
| 86P20 | 3758 a [†] | 0.34 a | 19.45 a | 3510 ab | 308 a | 1.155 a |
| AG1201 | 2281 c | 0.32 a | 17.71 a | 3230 ab | 310 a | 0.719 c |
| AG1203 | 3359 ab | 0.31 a | 16.41 a | 4044 a | 310 a | 1.022 ab |
| DKS37-07 | 3069 abc | 0.33 a | 17.34 a | 4352 a | 308 a | 0.95 ab |
| SP31A15 | 2757 bc | 0.29 a | 18.34 a | 2652 a | 297 a | 0.880 bc |
| Mean | 3027 B [‡] | 0.32 B | 17.87 B | 3401 A | 305 B | 0.939 B |
| **2018** | | | | | | |
| 86P20 | 3454 b | 0.53 a | 22.98 a | 2405 b | 331 a | 1.056 ab |
| AG1201 | 3357 b | 0.54 a | 22.48 a | 2712 ab | 333 a | 1.009 ab |
| AG1203 | 4173 a | 0.48 bc | 20.92 b | 3272 a | 331 a | 1.279 a |
| DKS37-07 | 3702 ab | 0.45 c | 23.33 a | 2485 b | 336 a | 1.100 ab |
| SP31A15 | 2687 c | 0.51 ab | 23.04 a | 2444 b | 335 a | 0.802 c |
| Mean | 3475 A | 0.50 A | 22.55 A | 2664 B | 333 A | 1.049 A |

[†] Within each column in each year, means with the same lowercase letter were not significantly different at $p$ = 0.05. [‡] In each column comparing two years, means with the same uppercase letter were not significantly different at $p$ = 0.05.

**Table 3.** Means of grain yield, harvest index, thousand kernel weight (TKW), seeds per plant, evapotranspiration (ET), and water use efficiency (WUE) across 2017 and 2018 as affected by planting date and hybrid interaction.

| Effect | Grain Yield kg ha$^{-1}$ | Harvest Index | TKW g | Seeds Per Plant | ET mm | WUE kg m$^{-3}$ |
|---|---|---|---|---|---|---|
| **PD1** | | | | | | |
| 86P20 | 2617 a [†] | 0.40 ab | 20.3 b | 1815 a | 300 a | 0.862 a |
| AG1201 | 2367 a | 0.44 a | 22.0 a | 2032 a | 291 a | 0.792 a |
| AG1203 | 2838 a | 0.39 ab | 18.0 c | 3037 a | 296 a | 0.943 a |
| DKS37-07 | 2404 a | 0.37 b | 21.0 ab | 2099 a | 291 a | 0.812 a |
| SP31A15 | 2368 a | 0.38 ab | 21.6 ab | 1850 a | 299 a | 0.769 a |
| Mean | 2279 B [‡] | 0.37 B | 20.5 A | 2161 B | 286 B | 0.783 B |
| **PD2** | | | | | | |
| 86P20 | 4420 a | 0.50 a | 22.1 a | 3812 a | 341 a | 1.301 a |
| AG1201 | 4858 a | 0.47 b | 18.2 c | 3767 a | 350 a | 0.994 b |
| AG1203 | 3486 b | 0.44 b | 19.2 ab | 4047 a | 348 a | 1.410 a |
| DKS37-07 | 4493 a | 0.45 b | 19.6 ab | 4594 a | 357 a | 1.265 a |
| SP31A15 | 3062 b | 0.46 b | 19.7 ab | 3101 a | 340 a | 0.897 b |
| Mean | 4045 A | 0.46 A | 19.9 A | 3904 A | 349 A | 1.165 A |

[†] Within each column in each PD, means with the same lowercase letter were not significantly different at $p$ = 0.05. [‡] In each column comparing PD1 and PD2, means with the same uppercase letter were not significantly different at $p$ = 0.05.

### 3.5. Relationship between SCA and Grain Yield

In 2017 PD2, the SCA populations peaked at anthesis and declined to the soft dough stage, while in 2018 PD2, SCA populations peaked at anthesis and declined to the grain milk stage. In comparison, SCAs did not arrive until after grain sorghum in 2017 PD1 reached physiological maturity, and in 208 PD1, SCAs peaked at soft dough and declined to physiological maturity depending on the hybrid. A regression analysis for PD2 showed that there was a negative relationship between grain yield and SCA per leaf for both 2017 ($R^2$ = 0.42, $p$ = 0.00194) (Figure 4A) and 2018 ($R^2$ = 0.42, $p$ = 0.00426) (Figure 4B). In the PD1 of 2018, the SCA infestation was observed at the grain-filling stage and did not affect grain yield ($R^2$ = 0.02, $p$ = 0.4533; data not shown).

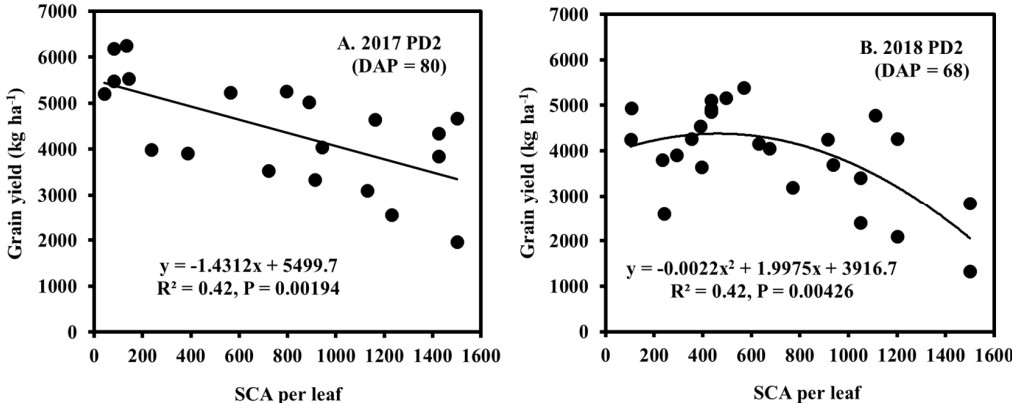

**Figure 4.** Relationship between SCA and grain yield for PD2 in 2017 (**A**) and 2018 (**B**). Each dot is the data of yield and SCA per leaf in each plot.

### 3.6. Evapotranspiration and Water Use Efficiency

Evapotranspiration was different ($p < 0.05$) between two growing seasons and the planting dates. There was a significant Y × PD interaction for ET ($p < 0.001$) (Table 2). However, there was no hybrid difference for ET. The mean ET was higher in 2018 (333 mm) than in 2017 (305 mm) and higher in PD2 (349 mm) than in PD1 (286 mm). WUE was influenced by all main factors and all two-way interactions except PD × H (Table 2). Similar to ET, plants in 2018 and PD2 had a higher WUE than those in 2017 or PD1. There were differences in WUE among hybrids in both years, which were averaged across planting dates. 86P20 (1.155 kg m$^{-3}$) in 2017 and AG1203 (1.279 kg m$^{-3}$) in 2018 had the highest WUE (Table 3).

### 4. Discussion

#### 4.1. SCA Infestation

In this study, SCA infestation occurred naturally and started in early September 2017 and the middle of July 2018. Apparently, environmental conditions largely affected the occurrence of SCA. The late SCA occurrence in 2017 could be related to the relatively lower temperature during the growing season as compared to the 2018 season. For both years, SCA infestation occurred mainly in PD2 plots. In 2018, temperatures dropped to 15 °C between 75 and 83 DAP, and it was observed that SCA populations quickly declined (Figure 3). It has been documented that SCAs are temperature sensitive, and population growth slows or declines when temperature reaches the range of 15–18 °C, but it is reported that this population drop is more likely due to dispersal than mortality [12,35,36].

In this study, early planting is a good cultural practice as compared to late planting if the goal is to reduce SCA infestation without the use of insecticide. However, grain yields were lower in PD1 than in PD2 in both years and for all hybrids (Table 3). Szezepanic (2018) reported similar results; there were approximately five fewer SCA per leaf in early planted grain sorghum (18 May) compared to for a more conventional planting date (8 June) in the Texas High Plains. Lipsey [37] also found a difference in SCA population between planting dates, where the early planting date also had a lower average SCA population. Studies from Szezepanic [28], Lipsey [37], and Pekarick [38] demonstrated that the SCA population varied among years and locations. These results indicated that planting date may not be an effective tool to minimize SCA population in every location or every year, which is related to the SCA migration patterns, as the SCA moving north varies year to year.

SCA population and timing of infestation influenced grain yield. In both years, PD1 had lower SCA populations during the later stages of reproductive development when compared to the PD2. There was no significant relationship between grain yield and SCA population in the PD1 of 2018 because SCA populations peaked later in the season at the soft dough stage. Balikai [39] found that SCA infestation after the milking stage did not affect grain yield. In 2018 PD2, SCAs had negatively impacted grain yield. This is

compared to infesting the later planting dates early in reproductive growth, anthesis to milking, showing that peak populations and differences in the timing of peak populations can result in different levels of grain yield reduction.

Hybrid selection still remains an important agronomic decision, since there were significant differences in SCA population among hybrids (Figures 1 and 2). Differences among hybrids were also found in grain yield and WUE (Tables 2 and 3). Hybrids showed different levels of tolerance to the aphids and demonstrated that SCA populations grew at different rates in different hybrids, and yield loss to SCA was higher in susceptible hybrid. SCA infestations can dramatically reduce grain yields from 30% to 100% in susceptible hybrids [11,12,16,40]. The differences in SCA population between susceptible and tolerant hybrids have been reported in previous studies, and tolerant hybrids often have lower average SCA populations and experienced less yield loss than susceptible hybrids [28,37,38]. The hybrid differences in the SCA population were not consistent across two years in this study. Because there was no SCA resistance at the gene level among the commercial hybrids in our study years, it is difficult to compare the results from different studies, particularly at different locations. At the same location, Szezepanic [28] showed that hybrid DKS37-07 was more tolerant to SCA than DKS44-20. In this study, DK37-07 showed more tolerance to SCA than SP31A15 in 2018 but showed no difference to SP31A15 in 2017. Nevertheless, the variations among hybrids in SCA infestation need more investigation in different environments.

### 4.2. Yield, Evapotranspiration (ET), and Water Use Efficiency (WUE)

The grain yield (1408–4938 kg ha$^{-1}$) and ET (240–374 mm) values reported in this study were within the range reported from a previous study at the same location. Baumdhart and Jones [41 reported dryland grain sorghum yield and ET in different tillage treatments from 1990 to 1995. Yield ranged from 470 to 5760 kg ha$^{-1}$, and ET ranged from 260 to 460 mm. However, WUE values (0.56–1.51 kg m$^{-3}$) in this study were greater than the values (0.18–1.25 kg m$^{-3}$) reported by Baumdhart and Jones [41]. For both years in this study, plants in PD2 (late June planting) performed better than those in PD1 (May planting) in yield, ET and WUE. Previous studies from the same location also demonstrated that later planting dates outperformed early planting dates in grain sorghum [20,42]. Allen and Musick [20] found that early planting dates (5 May) had lower grain yield and WUE than the medium planting date (23 May) and the late planting date (14 June) in different hybrids. Baumhardt and Howell [42] showed that a planting date of 5 June would yield approximately 5300 kg ha$^{-1}$ and would be significantly greater than the grain yield produced by a planting date of 15 May under dryland conditions. A more recent study showed that the highest median modeled grain yield in dryland grain sorghum resulted from the late June to early July planting date in the Texas High Plains [43].

In both years in this study, sorghum that had a late planting date received more precipitation and had greater seasonal ET than the sorghum that had an early planting date, supporting the development of seeds per plant and thus resulting in greater grain yield. Bell et al. [44] reported similar results whereby the seeds per panicle were increased by irrigating at a growing point differentiation. The authors also observed a positive relationship between the number of seeds/panicle$^{-1}$ and grain yield [44]. Even when the seasonal precipitations differences were less between two planting dates in 2018, plants in PD2 still had a higher grain yield, seeds per plant, HI, and WUE. In addition to receiving less precipitation, plants in PD1 in both years were subjected to different levels of hail damage, and the hail damage was more severe in 2017 than in 2018. In contrast, the PD2 sorghum in both years suffered less hail damage due to either being in early vegetative growth stages or having not emerged at all. This is in comparison to the PD1 sorghum that was often a week or two weeks away from entering the boot stage when hail damage occurred. In this study, hybrid differences in yield and WUE were only found in PD2, and SP31A15 had lower yield and WUE. This could be related to being relatively more susceptible to SCA infestation in this hybrid as compared to other hybrids.

## 5. Conclusions

SCA infestation was different among the hybrids, planting dates, and years. However, grain yields were lower for early planted sorghum than later planted sorghum. In the Texas High Plains, the early planting date and selection of SCA-tolerant hybrids can be considered effective methods for reducing SCA infestations if a producer is not using any insecticide, but producers should be aware that there may be a yield loss due to unfavorable weather conditions. However, yields across both years from PD2 without insecticidal control were 177% greater than yields from PDI. Increased grain yield would justify a later planting date over taking a yield drag to avoid SCA infestation. Nevertheless, planting date and hybrid selection are still the key cultural practices to avoid SCA populations and reduce the potential production risk of dryland grain sorghum.

**Author Contributions:** Conceptualization, Q.X. and J.M.B.; methodology, Q.X., Z.J., S.T., K.E.J. and J.M.B.; validation, Z.J., K.E.J., Q.X. and J.M.B.; formal analysis, S.T., Z.J., Q.X. and K.E.J.; investigation, Q.X., Z.J., S.T., K.E.J. and J.M.B.; data curation, Q.X., Z.J., S.T., K.E.J. and J.M.B.; writing—original draft preparation, Z.J., S.T.; writing—review and editing, Q.X., Z.J., S.T., J.M.B., B.C.B. and B.A.S.; supervision, Q.X., S.T. and K.E.J.; project administration, Q.X., J.M.B. and B.C.B.; funding acquisition, Q.X. and J.M.B. All authors have read and agreed to the published version of the manuscript.

**Funding:** This research was partially supported by the Texas Grain Sorghum Producers Board, Texas A&M AgriLife Research, and USDA-NIFA Hatch Project TEX09438.

**Data Availability Statement:** Data are available upon request.

**Conflicts of Interest:** The authors declare no conflict of interest.

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
