# Peer review of "Planting Date and Hybrid Affect Sugarcane Aphid Infestation, Yield, and Water Use Efficiency in Dryland Grain Sorghum"

_agronomy, doi:10.3390/agronomy12092033_

Round 1

Reviewer 1 Report

This article addresses current issues of great importance regarding planting date and Sugarcane aphid infestation on dryland grain sorghum for the High Plains region of Texas. However, some points of the article can be improved.

2 – I suggest that the title addresses only the Effect of planting date and sugarcane aphid infestation, for example, Effect of Planting Date and Sugarcane Aphid Infestation on Grain Yield of Dryland Grain Sorghum Hybrids in the High Plains region of Texas

96 – Was the year factor considered in the model? This must be described.

109-111 – The planting dates between 2017 and 2018 are different. The differences between PD1 and PD2 are more than a week. Is there any reason why this happened?

112 – The authors should better detail how the irrigation of the experiment was performed, in terms of periodicity.

114 – Authors must indicate the units of the traits evaluated.

115 – Authors must indicate the dates of SCA assessments.

125 – It would be interesting for the authors to present the mathematical formulas of ET and WUE.

149 – The differences in temperatures between the two years should be presented in the results and discussed.

164 – It would be more interesting for the authors to present a figure instead of table 1. In addition, should indicate in the figure the dates of establishment of the main phenological stages of the hybrids. This would make the results much more visual.

178-180 – It appears that the first-year SCA measurements were taken much later than the second-year ones, by about a month. The infestation peak occurred 68 days in 2018, a date not evaluated in 2017. How do the authors guarantee that the SCA infestation did not happen earlier in 2017?

195 – (Y x PD x H) is not a two-way.

199-201 – Does this statement refer to the two years or to 2018?

231 – What does it mean "...."?

264-270 – I expected the authors to do a more in-depth discussion on the influence of climatic effects on SCA infestation, mainly comparing assessment years and planting dates. However, only the temperature effect of a short period of 2018 was discussed. It would be interesting for the authors to expand on this discussion. What climatic effects can positively or negatively affect the incidence of SCA? At this point, I have doubts whether the differences in planting dates between the two assessment years and the late evaluations in 2017 were not the leading causes of the differences in this study. The authors need to clarify this.

279-280 – In my view, this statement is incorrect. The planting date is effective to minimize SCA population and the authors make this evident throughout the discussion.

328-340 – Authors should clearly state all reasons for reduced grain yield with early planting.

Author Response

The responses are in a Word file. Thank you. 

Reviewer 2 Report

Dear Authors,

Detailed notes on the manuscript are as follows:

1) Specify in the methodology why the agricultural research lasted 2 years (normally it should be a minimum period of 3 years)

2) The citations in the text do not comply with the MDPI standard (should be [no])

3) You used ANOVA + LSD in the data analysis, which indicates that the ANOVA was parametric (please attach the information that a test of the normality of the data population distribution was performed and the study of homogeneity of variance in samples, in the "results" section, enter the values ​​of these tests)

4) Figure 1. Average Sugarcane Aphid ... there is no need to re-inform about the "p" value - you provided it in the methodology, provide information about the error bars (Sd, Sd + mean,% ....), "A-c" are homogeneous groups

5) Figure 2. Average sugarcane…. The data on the charts should not be combined - this indicates the data prediction (leave a spread of points or add a trend line if you want to visualize the phenomenon, you can enter the value of the coefficient of determination, regression equation, etc.)

6) Tables not compliant with the MDPI standard (specify what the letters "a, A, b ..." mean)

7) Figure 3. Relationship between ... - explain why you used a linear trend in one figure and a 2nd order polynomial in the other

8) References - inconsistent with the MDPI standard

Author Response

The responses are in the Word file. 
